# Mitophagy Induced by Metal Nanoparticles for Cancer Treatment

**DOI:** 10.3390/pharmaceutics14112275

**Published:** 2022-10-24

**Authors:** Deepa Mundekkad, William C. Cho

**Affiliations:** 1Centre for NanoBioTechnology (CNBT), Vellore Institute of Technology, Vellore 632014, India; 2Department of Clinical Oncology, Queen Elizabeth Hospital, Kowloon, Hong Kong SAR, China

**Keywords:** metal oxide nanoparticles, mitophagy, dysfunctional mitochondria, cancer, oxidative stress-related pathway

## Abstract

Research on nanoparticles, especially metal nanoparticles, in cancer therapy is gaining momentum. The versatility and biocompatibility of metal nanoparticles make them ideal for various applications in cancer therapy. They can bring about apoptotic cell death in cancer cells. In addition to apoptosis, nanoparticles mediate a special type of autophagy facilitated through mitochondria called mitophagy. Interestingly, nanoparticles with antioxidant properties are capable of inducing mitophagy by altering the levels of reactive oxygen species and by influencing signaling pathways like PINK/Parkin pathway and P13K/Akt/mTOR pathway. The current review presents various roles of metal nanoparticles in inducing mitophagy in cancer cells. We envision this review sheds some light on the blind spots in the research related to mitophagy induced by nanoparticles for cancer treatment.

## 1. Introduction

Nanoparticles have wide applications in the medical field with important roles to play in cancer management. They help in imaging as contrast agents, drug carriers in gene delivery, etc. [1]. They are explicitly used in cancer therapy due to their capacity to deliver drugs to remote regions of the body that are normally inaccessible [2]. Among the different types of nanoparticles, metal nanoparticles are exceedingly used for medical applications due to their thermal, chemical, and optical features [3]. Compared to their counterparts, the metal nanoparticles possess many highly active uncoordinated sites with a large surface-to-volume ratio which makes them attractive. They are found to have a catalytic effect due to their special structural and physical features like the active surface atoms that can change size and shape rendering structural flexibility. The optical polarizability, electrical conductivity, ease with which they can complex with biopolymers, etc. make them ideal for biomedical applications [4] including cancer therapy.

Research reveals that the antitumor potency increases when conventional drugs are conjugated with nanoparticles. This is mainly due to the subcellular performance of nanoparticles to penetrate and reach organelles such as endosomes, nucleus, mitochondria, endoplasmic reticulum, Golgi apparatus, etc. Metal nanoparticles have a specific affinity towards mitochondria due to the membrane potential difference and can bring about the inhibition of mitochondrial respiration and can induce cell death in tumor cells by various mechanisms such as apoptosis and autophagy [5]. The role of nanoparticles in mitochondria-mediated cell death (mitophagy) is a relatively less exploited area of research. This review focuses on the role of metal-based nanoparticles in inducing mitophagy in tumor cells. 

Even though mitochondria are best known as ‘the powerhouse of the cell’, it possesses other important roles in cellular metabolism. Mitochondria are recognized as the center of oxidative phosphorylation producing highly reactive oxygen species (ROS). It is the converging point of many metabolic processes and is also actively involved in the cell cycle process, cell differentiation, and cell death. That is why mitochondrial dysfunction is involved in the proliferation and progression of cancer cells. Any dysfunction in the mitochondria can lead to the disruption of oxidative phosphorylation, resulting in reduced energy metabolism, ROS accumulation, inflammation, etc. which render cancer progression. Therefore, targeting mitochondria is an emerging trend in cancer therapy to induce apoptosis [6]. 

Molecular mechanisms such as autophagy and apoptosis take up the housekeeping role to help the cells to eliminate faulty organelles. Nanoparticles can act as autophagy modulators due to their effect on signaling pathways, thus creating an overstimulating signaling cascade in cancer cells than in non-cancerous cells. ROS-induced autophagic cell death brought about by silver nanoparticles in cancer cells was studied as a selective mechanism of autophagic cell death [7]. The unique property of nanoparticles to induce or inhibit ROS to induce toxicity in cancer cells has enabled their usage in the medical field. In normal skin cells treated with zinc oxide nanoparticles, abnormal accumulation of autophagosome was observed resulting from ROS accumulation in a concentration dependent manner. It was also found to activate autophagy through the inhibition of PI3K/Akt/mTOR signaling pathway [8]. The differences and similarities between autophagy and mitophagy are summarized in Table 1 and the various effects brought about by nanoparticles while inducing autophagy are listed in Table 2.

Mitophagy is the molecular event where dysfunctional or damaged mitochondria are effectively degraded and eliminated (Figure 1). Programmed mitophagy generally happens during the developmental process as a specialized form of autophagy. Persistent damage to mitochondria during stress and other pathophysiological conditions can lead to mitophagy. Starvation or hypoxia can also lead to mitophagy. Impaired mitophagy is common among autoimmune diseases, cardiovascular diseases, neurodegenerative diseases, metabolic disorders, and various types of cancers [29]. As metabolic reprogramming is common in cancer, the mitophagic processes are dysfunctional; mitophagy behaves as a tumor promotor or suppressor depending on the components, the type of cancer, and the microenvironment of the cancer cells. For example, BCl_2_ Interacting Protein 3 (BNIP3) is a pro-mitophagic receptor that is supposed to induce mitophagy. However, it functions as a tumor suppressor in breast cancer but acts as a promoter of tumor activities in the case of melanoma, renal cell carcinoma, and pancreatic cancer [30]. Other modulators or regulators of mitophagy such as PTEN-induced putative protein kinase 1 (PINK1) and Parkin, also have distinct roles in mitophagy. PINK1/Parkin has tumor suppressive activity and is frequently deleted in several types of cancers such as breast, ovarian, bladder, etc. [31] confirming that the loss of function of PINK1/Parkin can lead to the inhibition of mitophagy and in turn promote tumorigenesis in various types of tumors [32]. Overall, this indicates that mitophagy may have dual roles of inducing cell death as well as promoting cell survival [33]. 

Understanding the role of various components such as Pink1/Parkin, BNIP3, FUN14 domain-containing protein 1 (FUNDC1), optineurin (OPTN), microtubule-associated protein 1A/1B-light chain 3 (LC3), etc. in mitophagy will help in managing tumor progression. Chemotherapeutic drugs can induce cytotoxic effects through the induction of mitochondrial dysfunction as a result of oxidative stress that has an inhibitory effect on these components [33]. Many natural compounds are found to have varying effects in the induction of mitophagy [29]. However, at times, the complex nature of mitochondria makes it a challenge to target them for mitophagy. Consequently, a series of research have revealed that conjugating the drugs with ligands that target mitochondria can selectively perturb mitochondrial functions. Some of these studies used nanoparticles as effective drug carriers as they are very responsive to photosensitizers, radiosensitizers, and theranostic agents. They can target the energy machinery of tumor cells and effectively manipulate the underlying functional mechanisms in tumor cells. Nanoparticle-induced toxicity is an emerging area of research with special emphasis given to mitophagy as the means of cell death induced by nanoparticles [34] (Figure 2).

## 2. Different Metal-Based Nanoparticles and Their Mode of Action

Metal-based nanoparticles are used extensively in the medical field as drugs and imaging agents [35]. One way by which the nanoparticles exert their therapeutic effect is the covalent bonding with biomolecules. The covalent binding of metal-based nanoparticles with other biomolecules is responsible for the extensive ligand exchange chemistry of the drug. The ligand exchange property of nanoparticles helps to understand and interpret the molecular events at atomic levels so as to study the variations in the property of the drug. The formation of a covalent amide bond with carboxylic acid on the surface of a mesoporous silica nanoparticle, in synergy with zinc oxide quantum dots, was responsible for the inherent anticancer properties. Induction of mitophagy was found to be associated with the anticancer property triggered by excessive ROS formation [36]. The redox balance is important for cellular homeostasis that regulates a plethora of biological processes deciding the physiological well-being of the cell. Zinc oxide nanoparticle was found to induce ROS-mediated autophagy in CAL 27 oral cancer cell lines [37]. PINK1/Parkin-mediated mitophagy was reported as the basis for the anticancer activity of the metal oxide nanoparticle. The high expression of ROS in cancer cells results in the oxidation of cell components resulting in the loss of cell function and triggers autophagy. HC11 cells, treated with silver nanoparticle increased the expression of proteins associated with oxidative stress [38]. Hemeoxygnase-1 (HO-1), Kelch-like ECH-associated protein 1 (Kaep1), BTB and CNC homology proteins 1 (Bach1) and nuclear factor erythroid related factor 2 (Nrf2) were found to increase in the cells treated with the nanoparticles.

Another mode of action of nanoparticles is the activation of the immune system by phototherapy. Metal-based nanoparticles are excellent photosensitizers; they can be easily sensitized even by low-intensity light sources as in photodynamic therapy (PDT) and the irradiation creates ROS in cancer cells that can affect the mitochondrial membrane potential to induce mitophagy. In photothermal therapy (PTT), the nanoparticles sensitized with intense light and electromagnetic waves can instigate the expression of heat shock proteins (HSP) on the cell surface which simultaneously leads to the release of cytokines and other inflammatory regulators [2]. The immune system of the cells’ defense mechanism is activated upon the release of antigens in photo-immune therapy (PIT). The interventional PDT and PTT can trigger signaling molecules that act on the mitochondrial proteins directing the mitochondria for destruction by mitophagy. Iron oxide nanoparticle was directed to induce mitophagy in MCF-7 cell lines through a photothermal effect. When the photosensitized nanoparticles enter the cancer cells, they form aggregates in lysosomes and exocytosis is inhibited [39]. 

## 3. Selectivity and Targeting of Mitophagy by Nanoparticles

Early reports on the depolarized mitochondria being deported for damage control indicated mitochondrial autophagy through selective targeting. The involvement of specific proteins such as the key component ATG32 and BNIP3L/NIX, DRP1, etc. were identified and found to have a decisive role in selective targeting [40]. As a sign of cell survival, this mechanism is responsible for clearing the damaged, superfluous, or aged mitochondria with a definite advantage of complete turnover of all the components including the membrane and its associated proteins. Mitochondrial degradation needs to be highly specific and selective as even in extreme conditions such as natural or induced starvation, the mitochondria should be preserved as a source of energy. In such cases, restricted mitochondrial fission followed by a fused mitochondrial network will prevent mitophagy. The bite-sized fragments formed because of this selective induction of mitophagy are degraded. Thus, selective mitophagy throughout the prolonged starvation period is an adaptive response by the cell’s defense mechanism to augment and optimize the mitochondrial population. Activation of mitophagy and blockage of mass autophagy was observed in human hepatocellular carcinoma (HepG2) cells treated with silver nanoparticles. Mitochondrial fission induced by Drp1 and oxidative stress promoted mitochondrial degradation but blocked autophagic flux [41]. Similar results were observed in NSCLC cell lines when gold nanoparticle promoted Drp1-dependent mitophagy activation [42]. In addition to the mechanism of action, the size, shape, and conjugation of the nanoparticles with other biomolecules also decide the selectivity of mitochondria in nanoparticle-induced mitophagy. The effective strategy is to conjugate nanoparticles with peptides and amino acids to effectively deliver them to the receptors on the mitochondrial membrane and make them ready for mitophagy rather than any other type of programmed cell death (such as apoptosis or necrosis). However, understanding the relationship between the size, shape and conjugation of nanoparticles will help to target their interactions as it acts as a feedback mechanism for inducing mitophagy. Furthermore, these interactions are believed to have a decisive role in the distribution and intracellular trafficking behaviors of nanoparticles in terms of mitophagy [43]. The feedback mechanisms for the initiation of mitophagy demand autophagosome sequestration and nanoparticles can be selectively manipulated to induce or inhibit mitophagy depending on how this differential role is required to affect the treatment outcome of various cancers [7]. The range of structural diversity of nanoparticles provides them with unlimited combinations that can be employed in the targeted delivery of cancer-specific drugs to mitochondria to effect mitophagy. Surface coating of the nanoparticles such as iron oxide with dimercaptosuccinic acid (DMSA), or 3-aminopropyl-trietoxysilane (APS) is found to affect the intracellular trafficking of drugs [44]. These mechanisms will help in directing the nanoparticles to induce mitophagy. 

Mitochondrial targeting can be active or passive. Active targeting is achieved by surface functionalization of the nanoparticles with mitochondria-specific ligands. These ligands could be special moieties that can either be loaded onto the drug or can be standalone. Ligands such as natural products (glycyrrhetinic acid), mitochondrial peptides, 𝛼-tocopherol, etc. can be used for active targeting of nanoparticles to the mitochondrial membranes. However, the limitations such as immunogenicity, high production costs, complexities during synthesis, off-target toxicity, prolonged blood circulation period, and delay in clearance from the biological system must be taken into account when designing the proper nanoparticle for active targeting. Passive targeting depends on the physiological and chemical microenvironment of the mitochondria. pH, surface charge, the potential difference between both membranes, surface functionalization, etc. are factors affecting the passive targeting of nanoparticles to mitochondria. This offers an advantage compared to active targeting as the flexibility of nanoparticles allows the manipulation of their physical properties according to the requirements. However, the disadvantage is the possibility of the formation of nanoparticle aggregates resulting in rapid clearance from the host system. 

## 4. Metal Nanoparticles and Their Roles in Mitophagy

In contrast with bulk autophagy, selective autophagy (like mitophagy) identifies specific organelles for degradation depending on cargo-specific receptor proteins. These receptor proteins act as chaperons for translocators to induce the effect and nanoparticles are found to be good candidates for inducing mitophagy due to their physiochemical and biochemical aspects [34]. Even though toxicity is a limiting factor in the extensive use of nanoparticles in medicine, there is research describing nanoparticle-induced toxicity being redirected for inducing apoptosis, oxidative stress, autophagy and even mitophagy [45]. Most of these nanoparticles can effortlessly cross biological barriers and therefore can promote mitophagy. It is essentially due to the physiochemical and biological advantages of nanoparticles that they are capable of crossing biological barriers. Nanoparticles are known to have enhanced permeability and retention effect (EPR) that help them to easily accumulate in the permeable vasculature. The EPR effect along with the ability of nanoparticles to reach specific locations to release drugs in a controlled mode can enhance their therapeutic index. 

Pink/Parkin and BNIP3 are the two major pathways involved in mitophagy. Several nanoparticles such as gold, iron, silver, zinc, etc., and their oxides can be found to induce mitophagy in cancer cells through various mechanisms involving these pathways (Figure 3). Some of the nanoparticles or their oxide forms are capable of inducing mitophagy by disturbing the membrane potential of the mitochondria, or increasing the ROS content in the cell, or influencing the signaling pathway [46]. They are even capable of acting as tracking molecules that can trace the routes of various components of mitophagy. 

### 4.1. Gold-Based Nanoparticles

Gold nanoparticles (GNP/Au NPs) are one of the most studied nanoparticles in cancer research. GNPs can be used to sensitize the tumor cells so that mitochondrial function can be altered accordingly. Human breast cancer cell lines (MDA-MB-231) incubated with GNPs were irradiated with 225 kVp X-rays and were found to influence mitochondrial function resulting in decreased cell survival. The GNPs induced oxidation in the mitochondrial membrane, and mitochondrial polarization was observed [47]. Fluorescently labeled GNPs (Cy5@ Au NPs) were found to have a high tolerance to lysosomal proteins whereby they could tolerate photobleaching and thus can be used for tracking lysosomes to image mitophagy [48]. Induction of mitophagy, concomitant with apoptosis, was observed in THP-1 cells exposed to gold nanoparticles. These functionalized nanoparticles affected oxidative phosphorylation and protein ubiquitination also. In another study, it was observed that the gold nanoparticle-peptide conjugate can induce mitophagy with a change in the mitochondrial membrane potential. This type of association with nanoparticles and peptides was found to assist cell metabolism as well, even as intracellular trafficking was activated [43]. Autophagic mitochondrial fission was observed in NSCLC cell line treated with AuNP. The AuNP was found to cause excessive mitochondrial fragmentation in the cells under study. This was accompanied by a drastic increase in the recruitment of dynamin-related protein 1 (Drp1), mitochondrial dysfunctions, and enhanced induction of autophagy [42].

### 4.2. Iron-Based Nanoparticles

Iron nanoparticles (FeNPs), especially superparamagnetic iron oxide nanoparticles (SPIO-NPs), are found to induce mitophagy in cancer cells. Iron oxide nanoparticles were traditionally made use of in magnetic resonance imaging where they act as contrast agents. SPIO-NPs were found to recruit PARKIN from the cytoplasm to mitochondria, mediated by the protein PINK-1 located on the outer mitochondrial membrane [45]. The ubiquitination of PARKIN makes them susceptible to degradation by lysosomes and thus, mitophagy is induced in the cells under study. The increased involvement of mitochondrial proteins LC3-II and p62 in cells treated with iron oxide nanoparticle are further proof of the execution of mitophagy. Furthermore, iron oxide nanoparticles exhibit enzyme mimicking properties [49] that could be translated to antitumor properties. The inherent enzyme-like activity of iron oxide nanoparticles could initiate mitophagy and protect the cell from oxidative damage mediated by ROS molecules [50]. The ultra-small size of the iron nanoparticles is favorable for their easy transport to mitochondria where they can induce mitophagy after compromising the integrity of the mitochondrial membrane [51]. Cellular internalization was maximized when spindle-shaped iron oxide nanoparticles were used as nano transducers in mitochondria [52]. 

### 4.3. Silver-Based Nanoparticles 

Silver nanoparticles (AgNPs) are known to cause damage to DNA mediated by oxidative stress and mitochondrial dysfunction leading to cell death [53]. Silver ions can attach to protein receptors on the cell surface and bring about the denaturation of proteins resulting in pores in the cell membrane. This can lead to the disparity of membrane potential in mitochondria. In the A549 cell line, excessive ROS production and oxidative imbalance due to AgNPs were found to induce autophagy. This led to mitophagy mediated by PINK1/Parkin pathway as evidenced by upregulation of LC3 II/I, p62 expressions. Mitochondrial membrane potential was reduced accompanied by the upregulation of caspases 3 and 9, cytochrome c and BAX/BCl_2_. Human hepatocellular carcinoma (HepG2) cells, when treated with AgNPs, were found to stimulate mitochondrial fission and oxidative stress. There was a crosstalk between dynamin-related protein 1 (DRP1)-dependent fission and oxidative stress that triggered the AgNP-mediated mitophagy [41,54]. AgNPs can also decrease the membrane potential of mitochondria to stimulate mitophagy and apoptosis as observed in glioma cells [55]. As potent ROS inducers, AgNPs contribute to autophagic flux through redox signaling that involves hypoxia-inducing factors such as HIF-α, thus triggering mitophagy. The combined effect of ionizing radiation and AgNP on a panel of lung cancer cell lines revealed a dose- and time-dependent increase in protein oxidation releasing mitochondrial ROS. The exposure was found to result in decreased cell proliferation and caused cell cycle arrest in the cells under study [56]. (3-(2,4-dioxocyclohexyl)propyl-Net_2_-Coumarin (DCP-Net_2_C)) is a probe developed to analyze the sulfenylated proteins in mitochondria that are affected by treatment with AgNPs [57].. Rather than acting on the mitochondrial membrane permeability transition pore proteins, the silver nanocrystals coated with bovine serum albumin are found to interact with the phospholipid bilayer to induce mitochondrial membrane permeability transition (MPT) resulting in rupture of the mitochondrial membrane [58]. 

### 4.4. Zinc-Based Nanoparticles

The treatment of human tongue cancer cells (CAL 27 cells) with zinc oxide nanoparticles (ZnO NPs) resulted in an increase in the non-functional swelling of mitochondria implying cellular damage to mitochondria resulting in mitophagy and cell death. Further to this, an increase in the intracellular levels of reactive oxygen species along with a decrease in mitochondrial potential was also found in the cells treated with ZnO NPs [37]. The transport of Parkin from the cytosol to the mitochondrial membrane of the treated cells implies the execution of mitophagy by ZnO NPs [59]. The upregulation of hypoxia-inducible factor 1-α (HIF-1 α) endorsed by the inhibition of prolyl hydrolase and ROS was explained as due to mitophagy induced by ZnO NPs [60]. The synergistic effect of ROS and Zn ions was specially assessed in the upregulation of HIF-1 α. The treatment of osteosarcoma cells with ZnO NPs also was found to result in mitophagy. As a result of mitophagy, the cell adhesion protein β-catenin was degraded, and tumor metastasis was impaired in the cells under study [61]. 

## 5. Contradiction Where Nanoparticle Treatment Inhibits Mitophagy Instead of Promoting Mitophagy

An interesting research reported a contradictory result where ZnO NP treatment caused aberrant expression of LAMP-2 that resulted in impaired autophagic flux and sequential dysfunctional mitophagy [62]. This resulted in the accumulation of damaged mitochondria and accumulated ROS that was in total disagreement with previous research where ZnO NP induced mitophagy. Furthermore, the intracellular ROS levels were found to be efficient in chemodynamic therapy where the iron-based nanocatalyst effectively inhibited the PINK1/Parkin-mediated mitophagy in endometrial cells [50].

### Probable Mechanism of Action of Nanoparticles That Is the Basis for the Contradictory Action

Instead of promoting mitophagy, nanoparticles are found to inhibit mitophagy in some cases. The contradictory action may be because of the variation in biological and physicochemical properties exhibited by the nanoparticles. Though there are limitations to the toxicological assessment of nanoparticles (for example, the inability to quantify the correct dose to get an optimum, quantifiable in vivo effect), the in vitro tests have helped to successfully evaluate the interaction of the nanoparticle with the cellular environment and come up with plausible explanations for the contradictory effect. ROS generation is one of the major reasons for mitophagy induction where membrane potential is affected. Nanoparticles can inhibit intracellular ROS generation thus leading to the inhibition of mitophagy. Platinum nanoparticles retained mitochondrial membrane potential thus inhibiting intracellular ROS in the human brain glioblastoma cancer cell line [63].

## 6. Pathways Involved in Nanoparticle-Mediated Mitophagy

Mitophagy being an evolutionarily conserved mechanism, the metabolic processes and the cell’s defense mechanisms make sure to accurately execute the process of mitophagy to eliminate the damaged mitochondria at the earliest through an interplay of different signaling pathways. It is therefore imperative to understand the components of the pathway that are involved in the process of mitophagy mediated by nanoparticles. The metal nanoparticles are known to have an intrinsic selectivity in activating the pathways leading to mitophagy in cancer, especially as compared to their normal counterparts [64]. Multiple pathways drives mitophagy mediated by nanoparticles, and they may be dependent on regulatory, and signaling pathways, with regular crosstalk between them [29] (Figure 4). 

### 6.1. PINK1/Parkin Pathway

One of the most significant pathways of mitophagy is the PINK1/Parkin pathway which is based on ubiquitin which proceeds to degrade the damaged mitochondria. PINK1 is the first protein to respond to the damage in mitochondria as it can easily sense mitochondrial transmembrane potential loss [65]. PINK1 belongs to the serine/threonine kinase family that is activated under mitochondrial damage. PINK1 is generally very stable in a normal state as they are cleaved by matrix processing peptidase (MPP), and Presenilins-associated rhomboid-like protein (PARL). The accumulation of cleaved PINK1 is prevented by translocating them back to the cytosol to be degraded by proteasomal enzymes [66,67]. However, the cleavage of PINK1 and its further translocation back to the cytosol is impaired upon depolarization due to the loss of transmembrane potential in damaged mitochondria. There is an upsurge in PINK1 followed by the phosphorylation of ubiquitin molecules at serine 65 on the outer mitochondrial membrane. PINK1 along with the phosphorylated ubiquitin then recruits Parkin from the cytosol to the outer mitochondrial membrane where it conjugates with the phosphorylated ubiquitin. Parkin is a cytosolic E3 ubiquitin ligase that promotes the degradation of the ubiquitinated protein. With the help of LC3-II, the ubiquitinated proteins on the outer mitochondrial membrane lead the damaged mitochondria towards the lysosome for destruction by proteasomal enzymes [68]. Other proteins such as NDP52 and optineurin are also involved in the stimulation of mitophagy but they are found to act independent of Parkin [69]. 

Advanced studies show that when hepatic cells were treated with superparamagnetic iron oxide nanoparticles (SPIONs), the immunofluorescent signals given out by PINK1 were increased along with high proportions of mitochondrial LC3-II and p62 [46]. This is indicative of the role of SPION in PINK1/Parkin-dependent ubiquitin-mediated mitophagy. Further, biogenically synthesized selenium nanoparticle (Se NP) tested on IPEC-J2 cells were found to abate the fusion of mitochondria and lysosome, reduce the overproduction of ROS and decrease the mitochondrial membrane potential (MMP). The PINK1 and Parkin expression in the cells were found to be down-regulated confirming mitophagy induction by Se NPs [70] and Au NPs [71]. 

### 6.2. P13K/Akt/mTOR Pathway

PI3K/Akt/mTOR is another major signaling pathway involved in mitophagy. Phosphatidylinositol 3-kinases (PI3Ks) are a group of enzyme transducers involved in a variety of cellular activities including the growth and proliferation of cancer cells. Akt (Protein kinase B) is a serine/threonine kinase involved in supervising the movement of parkin to the damaged mitochondria [72]. The mammalian target of rapamycin (mTOR) is another highly conserved serine/threonine protein kinase that is phosphorylated at Serine 473. While mTOR is important for the formation of autophagosomes, its inactivation is essential for autophagy because hyperactivity of mTOR is found to repress PINK1 expression and this, in turn, will decrease the translocation of Parkin to mitochondria [73]. The proliferation of A549 cell lines was found to be affected by blocking the PI3K/Akt/mTOR pathway heralded by autophagy [74] as this pathway is classified as a negative regulator of autophagosome formation [38]. The decreased levels of mitophagy markers were observed accompanied by the inhibition of the PI3K/Akt/mTOR pathway in the cultured glioblastoma multiforme (GBM) cells treated with solid lipid curcumin particles (SLCP). Low expression levels of mitophagy markers were found after treatment confirming the signaling pathway inhibition. MCF-7 breast cancer cell lines treated with gold nanocomplex were found to have differential expression patterns of the genes belonging to the PI3K/Akt pathway [75]. Forkhead Box O1 (FOXO1), the transcriptional factor that is a downstream target of the Akt signaling pathway, was found to be activated by the treatment with nanocomplex. Additional data suggests the suppression of TSC2, a potent inhibitor of mTOR. The study concluded that the crosstalk between PI3K/Akt/mTOR pathways was essential to mediate the various mechanisms involved in the multiple pathways to induce the inhibitory effect on the cancer cell lines under study. mTOR also acts as a mediator in the crosstalk between PI3K/Akt and AMPK pathways and in PC-3 prostate cancer cells, Ag NPs were found to activate autophagy through the AMPK-mTOR pathway [14].

### 6.3. MAPK/ERK Pathway

MAPK/ERK pathway is a prominent signaling pathway involving mitophagy. Mitogen-activated protein kinase (MAPK) belongs to the line of ubiquitous proline-directed, protein-serine/threonine kinases. They are actively involved in the three-tiered protein kinase cascade controlling some of the major cellular activities leading to the functional and developmental organization of cells in an organism [76]. The extracellular signal-regulated kinase 1/2 (ERK 1/2), c-Jun N-terminal kinase (JNK) and p38 are the major responders to this pathway. While JNK and p38 MAPK pathways are related to stress-mediated apoptosis in cells, the MAPK/ERK pathway is fundamental in the signal transduction network [77]. The anti-apoptotic effect of the signaling pathway has a major role in cancer cell proliferation. Additionally, the MAPK/ERK pathway is found activated in mitophagy in response to oxidative stress [78]. Reports confirm that the activity-based localization of ERK2 to mitochondria is sufficient to induce mitophagy. The constitutive overexpression of active ERK2 further increased fusion proteins such as GLP LC3, a mitochondrial marker for autophagic vesicles, on the mitochondrial membrane emphasizing the overall role of this signaling pathway in mitophagy. The changes in MAPK/ERK pathway are highly dysregulated in malignant tumors. 

Sonodynamic therapy based on nano-sensitized liposomes showed a visible increase in MAPK/p38 phosphorylation regulated by ROS formation. An increase in phosphorylation suggested aggravated oxidative stress, and reduced mitophagic vacuolization with impaired Parkin translocation [79]. Absorption of iron oxide nanoparticles led to the failure of respiration and mitophagy of the cells. Gold nanoparticles, on the other hand, were found to induce autophagic flux resulting in impaired lysosomal function. 

### 6.4. Hypoxia-Inducible Factor-1α (HIF-1α) Pathway

There is a high demand for oxygen and nutrients in cancer cells; the failure of balancing the demands will lead to a hypoxic microenvironment in the tumor cells. Tumor cells counter this abnormality by modifying the expression pattern of transcriptional factors that respond to hypoxia. HIF-1 is the major component of the transcriptionally regulated pathway where under normal conditions, the prolines of the HIF-1α subunit become activated and interact with The von Hippel-Lindau (VHL) protein and are degraded. Under conditions of hypoxia, the hydroxylases of proline remain inactive and fail to form the complex with VHL. The unhydroxylated 1α subunit moves freely to the nucleus to associate with the β subunit (HIF-1β) and CBP/p300. This complex then combines with the hypoxia response elements (HRE). The association of the HIF-1β/CBP/p300 complex with HRE elicits more transcriptional activities resulting in the transcription of more genes downstream of the pathway. More than 60 such genes are known to be transcribed during the process. Many cellular metabolic activities are affected by the regulation of these processes [80]. Nanoparticles are known to be hypoxia-responsive once they extravasate into the tumor microenvironment. This quality promotes the efficiency of nanoparticles in tumor therapy [81]. Iron oxide nanoparticle-doxorubicin complexed with hypoxic cell radiosensitizer SAN (sanazole) induced downregulation of HIF-1α [82]. The associated genes, vascular endothelial growth factor (VEGF), and Akt are also downregulated. On the other hand, the treatment of A549 cells with AgNPs showed an increase in the expression of HIF-1α where exposure of the cells to hypoxia blocked oxidative stress induced by the nanoparticles. It was noted that the autophagic flux was restricted through the regulation of LC3-II and p62 [83]. 

### 6.5. Oxidative Stress-Related Pathways

It is well known that levels of ROS can decide the fate of cells; high ROS can lead the cells to apoptotic cell death whereas a low concentration of ROS in cells can lead to mitochondrial dysfunction. This can be a trigger for mitophagy where the damaged mitochondria are removed for cell survival and maintenance of cell homeostasis [84]. The treatment of cells with copper oxide (CuO) NPs aggravates the build-up of excessive ROS in the form of superoxide anions that may result in impaired mitophagic flux. Dysfunctional mitochondria are believed to be the source of ROS accumulation here. AgNPs were capable of regulating autophagy mediated by injury to mitochondria and lysosomes in A549 cells. Excessive ROS production with an imbalance between the oxidant/ antioxidant systems was evident in the cells under study [53]. 

## 7. Blind Spots in Research Involving Mitophagy, Nanoparticles, and Cancer

Much research and data are found on the cause and effect of mitophagy in research related to neurodegenerative diseases such as Parkinson’s disease. The mitophagic pathway is extensively studied in these disease conditions. It is understandable as there is a direct involvement of PINK1/Parkin in neurodegeneration. However, the fact that there is a crosstalk between the proteins involved in the process of mitophagy in neurodegeneration, inflammation, immunomodulation, and cancer seems to be overlooked. Different cancers such as colon, liver and pancreatic cancers have a serious overlap between inflammation and immunomodulation. Receptor-mediated mitophagy can reveal a lot about the mechanism of action of nanoparticles that can help in developing the nanoparticles as an effective strategy to combat various types of cancer. Further, new models such as 3D cell culture that can mimic human physiology could be developed. The 3D models can emulate the in vivo model in a much better way than the pretend in vitro environment. Experimentation with the 3D models will largely enhance the possibilities for trying multiple target proteins to be studied at the same time. In addition, 3D models will help to minimize research with animal models thus avoiding ethical issues to a larger extent. There are many challenges in studies involving nanoparticle-induced mitophagy, some of them are listed in Figure 5. 

When the mitochondrial metabolism is targeted by the molecules, it can result in the inhibition of the glycolysis/TCA cycle, redox signaling, or one-carbon metabolism responsible for the production of ROS and antioxidants. Proteins such as SOD, NADH, αTOS, etc. can also be targeted by these molecules resulting in the up- or down-regulation of ROS and other antioxidant enzymes such as GPX (Figure 6).

## 8. Recent Developments in Nanoparticle-Mediated Mitophagy

Through the advancement of the nanobiotechnology, scientists can fabricate new, sophisticated biomaterials that incorporate multiple functions and activities, and can offer a versatile platform for the newly fabricated materials [85]. Enzyme-instructed self-assembly (EISA) and aggregation-induced emission (AIE) are two advanced models employed in cancer therapy. 

### 8.1. Enzyme-Instructed Self-Assembly or EISA

EISA is a process where the self-assembly of cellular components such as protein is mediated by enzymatic processes. Molecular assemblies can be manipulated and modified to stimulate various enzymatic reactions leading to the self-assembly of peptides to form nanostructures. EISA substrates can form supramolecular assemblies that can selectively target cancer cells because of their high penetrating power. They can accumulate in the mitochondrial matrix and induce mitochondrial dysfunction leading to the initiation of mitophagy [86]. Nanofibers, formed by EISA, can be transported to mitochondria where they induce mitochondrial dysfunction even at a relatively lesser concentration. This could lead to cell death activated through multiple signaling pathways [87]. Self-assembled nano peptides could be directed towards PD-L1 (programmed cell death ligand 1) on the cell surface to selectively degrade and thus manipulate their levels so as to avoid immune escape. This high efficiency of nano peptides to bind to immune cells and manipulate them was probably due to the multivalent binding sites found on the surface of the self-assembled nano peptides [88]. Mitochondria-targeted EISA can serve as an alternate strategy to target cancer cells through the production of highly selective, multitargeted nano peptides with minimal drug resistance [89].

### 8.2. Aggregation-Induced Emission (AIE)

Certain fluorescent molecules can emit higher fluorescence upon entering a crystalline state. Organic compounds with luminescent properties show higher efficiency when aggregated as compared to the solution. This phenomenon called aggregation-induced emission is found to be very high in metal nanoclusters and helps them to locate proteins. Aggregation-induced emission probes are developed that are sensitive to viscosity and regardless of the intensity of the mitochondrial membrane potential, this probe can be discreetly directed to the mitochondria. This is a unique but accurate way of detecting mitophagy [90]. The real-time monitoring of mitochondrial viscosity proved to be better as it is closely associated with the mitochondrial respiratory state reflecting the state of physical wellness or disease of the mitochondria. A near-infrared fluorophore with lipocationic property was developed to selectively accumulate in the mitochondria of cancer cells [91]. This competent multimodal theranostic agent could evaluate mitophagy activities through theranostic approaches. These nanoaggregates could induce mitophagy and block mitophagic flux to accelerate apoptosis in cancer. They were found to be highly beneficial in tracing mitophagy in apoptotic cells in PDT [92,93]. Combining the optoelectronic and sensory properties of these aggregates, they have wide applications in imaging and theranostics. The strong, dynamic intermolecular interactions, polarity, and the power of emitting light confer many more possible roles to the AIE system for cancer therapy.

## 9. Mechanistic Role of Anti-Tumor Nanoparticles in Inducing Mitophagy

Autophagy is a survival mechanism adopted by cells. In cancer cells, it can be activated to mediate resistance to chemotherapeutic agents. This spontaneous resistance can interfere with the efficacy of the therapeutic agent. However, there are very less interventions to mechanistically regulate autophagy. This is because the drugs that can inhibit or activate autophagy (such as rapamycin and hydroxychloroquine) have not been developed properly for this function [39]. It was also observed that the intrinsic pharmacological specificity of these drugs is very low to explicitly target the components of autophagy. This problem related to specificity arises from the structural complexity of tissues, the wide range of homologous and heterologous interactions that they are subjected to, and due to their failure to specifically target a single cell type. However, these challenges may be overcome by the nanoparticles, at which nanoparticles can explicitly target the decisive molecule in the pathway. Additionally, enhanced efficacy, stability, solubility, and adaptability make them pharmacologically effective in acting as anti-tumor agents. 

Increased expression of Beclin-1, XBP1, CHOP, and LC3II in cancer cells suggested the AgNPs-induced mitophagy in cancer cells. The downregulation of ATG3 and ATG12 was also observed in AgNP-treated cells [94]. The drug-resistant gene, NPRL2, showed increased expression in cancer cells and upon upregulation, it repressed the mTOR signaling pathway to activate the process of autophagy and suppressed apoptosis. Table 3 explains the role of some of these proteins in the induction of mitophagy. The ROS-scavenging properties of nanoparticles such as iron oxide (Fe_3_O_4_) [95], ZnO, and silica (Si) [96] nanoparticles can disrupt the antioxidant mechanism of the cell resulting in oxidative imbalance, finally inducing stress-related mitophagy [96,97,98]. The exposure to graphene oxide (GO) leads to the disruption of autophagic flux and weakening of lysosomal function resulting in the accumulation of the substrate for autophagy such as the autophagosome cargo protein p62 also called sequestosome-1 (p62/SQSTM) [99]. This can lead to mitochondria-mediated apoptosis in the cancer cells. Fe_3_O_4_ NPs are also found to induce excessive autophagy leading to endothelial dysfunction and inflammation that is assumed to be associated with the Beclin-1/VPS34 complex [100].

## 10. Conclusions

Beyond protein and metal ion corona that can act as probable targets, high-throughput data can be generated to study the behavior of the nanoparticle when exposed to different biological conditions. This will give clarity to the dose-time effect in relation to nanoparticle toxicity. The integration of automated platforms (such as high-throughput screening) and ‘-omics’ technologies could be an effective strategy to fill the existing knowledge gap in nanoparticle-induced mitophagy in cancer. It is expected that the technical advancements in the field of nanotechnology will inspire the scientific community to integrate these techniques (such as EISA) and develop a multi-disciplinary approach to cancer therapy. 

## Figures and Tables

**Figure 1 pharmaceutics-14-02275-f001:**
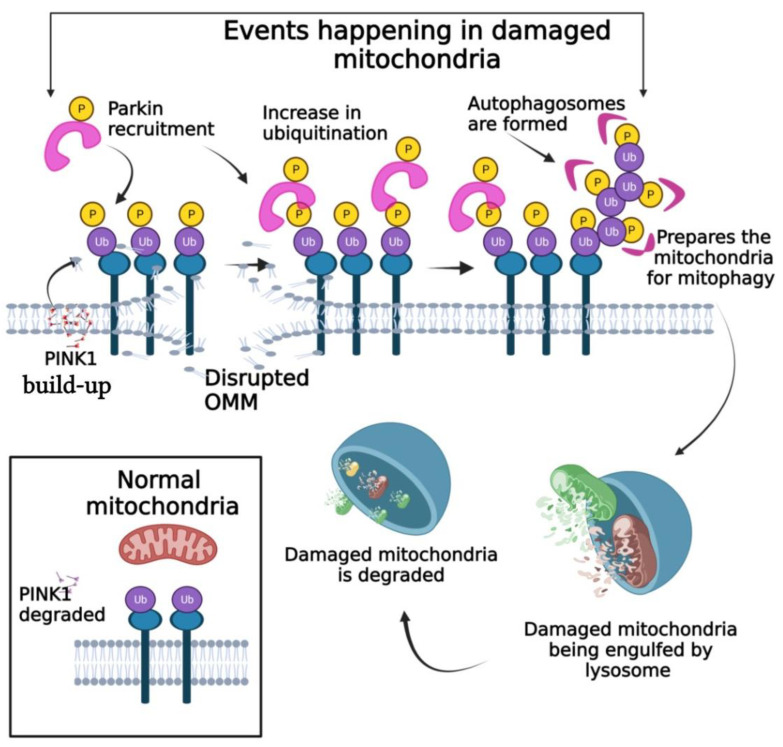
The process of mitophagy. When the outer mitochondrial membrane is disrupted, PINK1 is accumulated which in turn helps in the recruitment of Parkin and other receptor proteins. Autophagosomes formed prepare the mitochondria for mitophagy where the damaged or dysfunctional mitochondria are engulfed by lysosomes and degraded.

**Figure 2 pharmaceutics-14-02275-f002:**
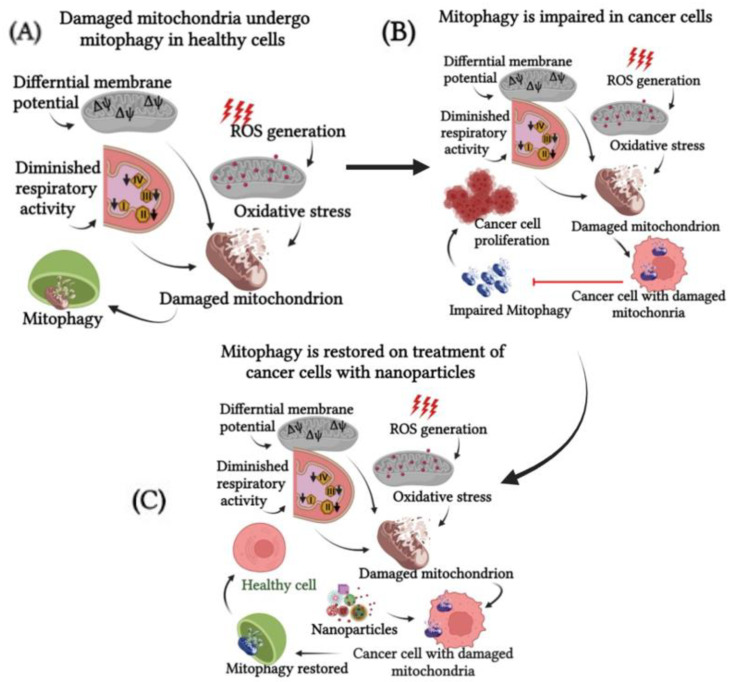
Effect of nanoparticle on mitophagy in cancer cells. Healthy cells undergo mitophagy (**A**) whereas cancer cells accumulate damaged mitochondria and mitophagy is impaired (**B**). Nanoparticles can restore normal homeostasis by restoring mitophagy and thus protecting the cells (**C**).

**Figure 3 pharmaceutics-14-02275-f003:**
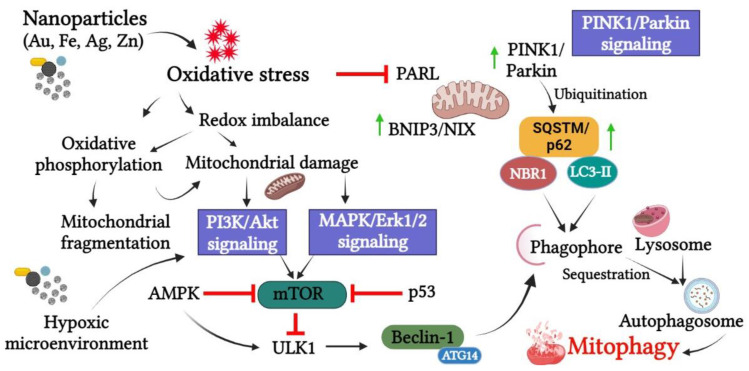
Major molecular pathways implicated in mitophagy that is induced by nanoparticles. Au—gold, Fe—iron, Ag—silver, Zn—zinc, PARL—presenilin-associated rhomboid-like protein, BNIP3—BCL2/adenovirus E1B 19 kDa protein-interacting protein 3, NBR1—neighbor of BRCA1, LC3-II—microtubule-associated protein 1A/1B-light chain 3, AMPK—AMP-activated protein kinase, PI3K—phosphatidylinositol-3-kinase, Akt—protein kinase B, MAPK—mitogen activated protein kinase, Erk 1/2—extracellular signal-regulated kinase 1/2, AMPK—AMP-activated protein kinase, mTOR—mammalian target of rapamycin, ULK1—Unc-51–like kinase 1, ATG14—autophagy-related 14.

**Figure 4 pharmaceutics-14-02275-f004:**
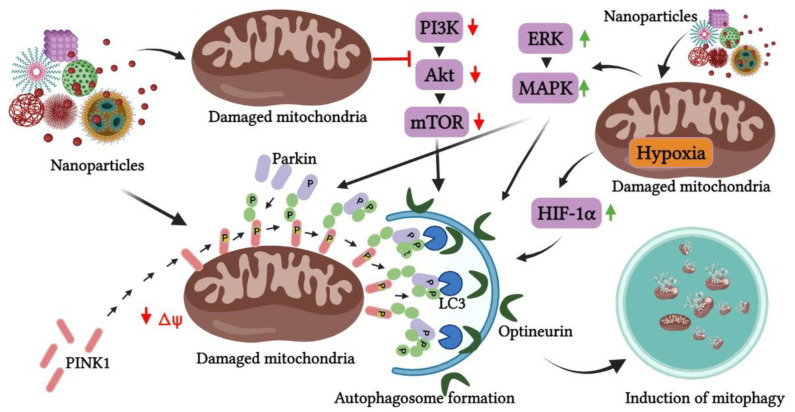
Crosstalk between different signaling pathways in nanoparticle-induced mitophagy. The involvement of nanoparticles in the induction of mitophagy is regulated mainly by PINK1/Parkin pathway with the involvement of additional pathways such as PI3/Akt/mTOR, ERK/MAPK and HIF-1α. (PI3K—phosphatidylinositol 3-kinase, Akt—protein kinase B, mTOR—mammalian target of rapamycin, ERK—extracellular signal-regulated kinase, MAPK—mitogen-activated protein kinase, HIF-1α—hypoxia-inducible factor 1 alpha, LC3—microtubule-associated protein 1A/1B-light chain 3).

**Figure 5 pharmaceutics-14-02275-f005:**
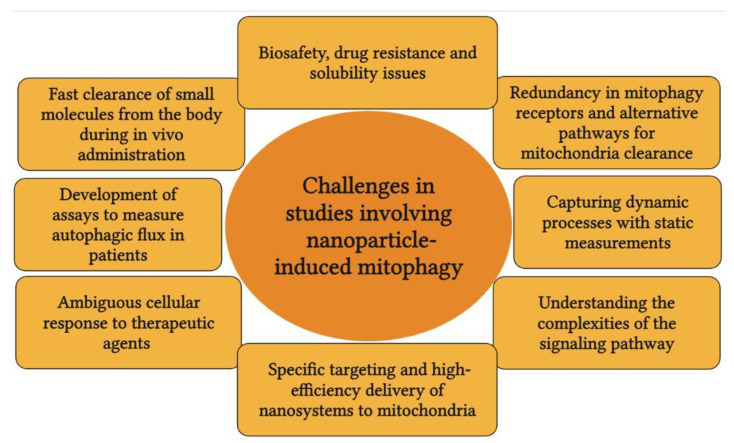
Challenges in studies involving nanoparticle-induced mitophagy.

**Figure 6 pharmaceutics-14-02275-f006:**
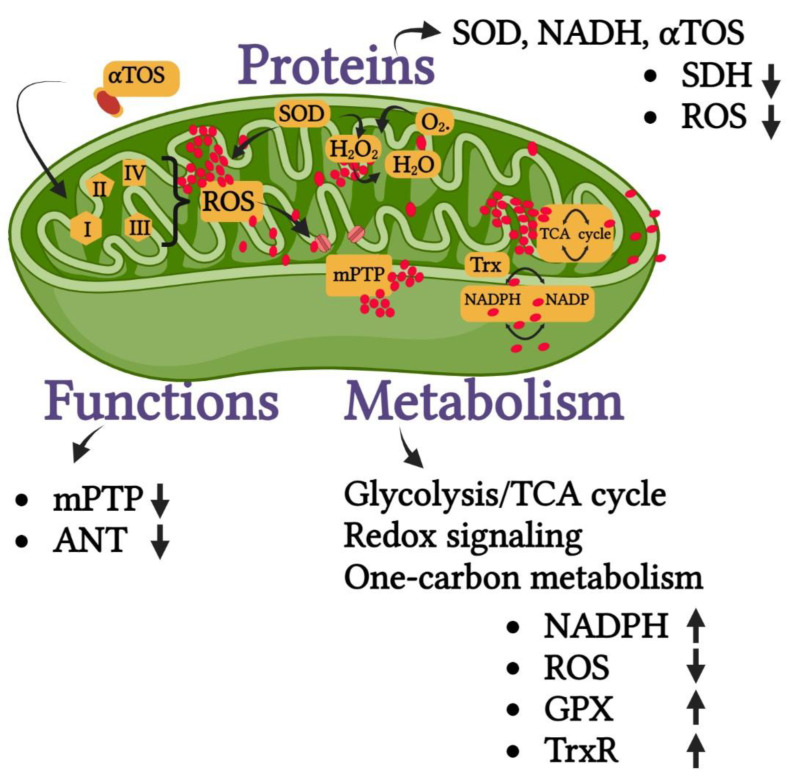
Possible targets to induce mitophagy in cancer. In addition to mitochondrial proteins, the function and metabolism of the mitochondria may be targeted with the metal nanoparticles resulting in mitophagy. The increase in the expression of the reaction product is indicated by the up arrow and the decrease is indicated by the down arrow. SOD—superoxide dismutase, NADH—nicotinamide adenine dinucleotide, αTOS—alpha-tocopherol succinate, SDH—succinate dehydrogenase, ROS—reactive oxygen species, mPTP—mitochondrial permeability transition pore, ANT—adenine nucleotide translocase, TCA—tricarboxylic acid, NAPDH—nicotinamide adenine dinucleotide phosphate, GPX—glutathione peroxidase, TrxR—thioredoxin reductase.

**Table 1 pharmaceutics-14-02275-t001:** Differences and similarities between autophagy and mitophagy.

	**Autophagy**	**Mitophagy**
Type	General form of degradation of cellular components including organelles	Specific degradation of mitochondria
Regulation	Dependent on the nutrient/energy/stress signals	Independent of the nutrient/energy/stress signals
Stimulus	Nutrient and energy stress, ER stress, pathogen-associated molecular patterns (PAMPs), danger-associated molecular patterns (DAMPs), hypoxia, redox stress, mitochondrial damage	Mitochondrial membrane depolarization, changes in cytosolic pH
Malleability	Malleable as it can degrade specific targets, entire organelles, and large portions of cytoplasm	Not malleable at all as it degrades only mitochondria
Substrate	Sequestosome 1/p62 (SQSTM/p62)	Mitophagy involves unique and additional substrate identification mechanisms, notably PTEN-induced kinase 1 (PINK1) and parkin RBR E3 ubiquitin protein ligase (PRKN)
Types	Three types— Chaperone-mediated autophagy (initiated by chaperone Hsc70 and recognizes one protein at a time), Microautophagy (initiated by invagination of lysosomal membranes. Lipid, protein or organelles can be degraded) and Macroautophagy (double membraned organelles are degraded)	Only one type where Only mitochondria are degraded
Methods of detection	Conventional electron microscopy to detect the autophagosome number and autophagic flux Fluorescence microscopy to count the average number of punctate structures per cell (puncta formation assay) Immunoblotting to detect the conversion of the cytosolic form of LC3 (LC3-1) to its membrane-bound lipidated form (LC3-II) Flow cytometry to detect the degradation of autophagy-selective substrates	Electron microscopy Fluorescence microscopy to detect co-localization of mitochondria with autophagosomes or lysosomes Western blotting to measure the degradation of mitochondrial proteins (like LC3) Fluorescent protein-tagged assays like MitoTimer, mt-Keima, and mito-QC
**Similarities**		
Activation of similar molecular machinery (i.e., BECN1, ULK1/2, LC3)		
Generation of ROS molecules can act as a trigger for both processes		
Both processes result in cellular damage and apoptosis		
Essential for quality control of the cells’ defense mechanism		
Both processes are forms of degradation mechanisms rather than cell death mechanisms (unlike apoptosis).		

SQSTM1—Sequestosome 1, BECN1—the gene encoding the protein Beclin 1, ULK 1/2—Unc-51 such as autophagy activating kinase (1/2), LC3—Microtubule-associated protein 1A/1B-light chain 3.

**Table 2 pharmaceutics-14-02275-t002:** Autophagy induced by different nanoparticles.

Nanoparticle (Size)	Mode of Cellular Recycling	Mechanism/Outcome	Conjugation	In Vitro/In Vivo Model	References
Polydopamine nanoparticle (101.96 ± 6.70 nm)	Autophagy	Photothermal cell killing	Beclin 1- derived peptide (Beclin 1), polyethylene glycol (PEG) and cyclic Arg-Gly-Asp (RGD) peptides (PPBR)	NIH3T3 cells, HeLa cells	[9]
AuNPs (15 nm)	Autophagy	ROS generation by cellular uptake	Poly (acryloyl-L,D) and racemic valine (PAV)	MDA-MB-231 cells	[10]
Autophagy Cascade Amplification NPs (Self-assembled peptide-cholesterol monomers) (150 nm)	Autophagy	Overactivated autophagy and enhanced tumor antigen processing	Oxaliplatin prodrug (HA-OXA)	CT26 tumor-bearing mice	[11]
ZnO NP (300 nm)	Autophagy	ROS-mediated enhanced tumor chemotherapy by overstimulated autophagy	Bare	P-gp-mediated multi-drug resistant human breast cancer cells (MCF-7/ADR cells), BALB/c mice	[12]
Nickel oxide NPs (24.05 ± 2.9 nm)	Autophagy	Oxidative stress, JNK activation	Bare	HeLa	[13]
Silver NPs (78 nm)	Autophagy	Lysosome injury and cell hypoxia	Bare	Prostate cancer cell line (PC-3)	[14]
Silver NPs (11–23 nm)	Enhanced autophagy	Inhibition of NLRP3 inflammasome	Bare	THP-1 cells, AMJ-13 cells, HBL cells	[15]
Gold NPs (60.00 ± 4.24 nm)	Autophagy	ROS-mediated cell death	Bare	Human ovarian adenocarcinoma cells (SKOV-3)	[16]
Hollow mesoporous titanium dioxide nanoparticles (HMTNPs) (∼100 nm)	Escape from macrophage phagocytosis	Sonodynamic Therapy	Hydroxychloroquine sulphate (HCQ)	MCF-7, MDA-MB-231, HepG2, NIH3T3,	[17]
Copper oxide NPs (5.4 ± 1.2 nm)	Destructive autophagy	Enhanced radio-sensitizing effect	Bare	MCF-7	[18]
ZIF-82-PVP nanocrystals (~240 nm)	Autophagy	Apoptosis promoted by X-ray-induced nitrosative stress	Conjugation with PVP	MDA-MB-231, 4T1 and Panc-1 cells	[19]
Fe_3_O_4_ NP (26.3 ± 4.42 nm)	Autophagy	ROS-mediated NF-κB and TGF-β signaling pathway activation	Polyethyleneimine	HeLa	[20]
Branched Au-Ag NPs (~200 nm)	Autophagy	Photothermal toxicity	Polydopamine-coated	Human bladder cancer cells (T24 cells)	[21]
TiO_2_ NPs (20–30 nm)	Autophagy blockage	Cytotoxicity and apoptosis induction with enhanced chemotherapeutic effect	5-fluorouracil	Human AGS gastric cancer cells	[22]
Au NPs (42.6 ± 5.3 nm)	Autophagy	Immunogenic cancer cell death	Polydopamine	MCF-7 and MDA-MB231	[23]
Copper-palladium alloy tetrapod nanoparticle (~50 nm)	Pro-survival autophagy	TNP-1-mediated photothermal therapy	-	Triple-negative (4T1), drug-resistant (MCF-7/MDR) and patient-derived breast cancer models	[24]
Spheroid fluorescent polystyrene nanoparticles (PS-NPs) (30 nm)	Selective autophagy	Inhibition of autophagosome formation and rescued ATG4-mediated autophagy	Functionalized with amino groups	OAW42 cells	[25]
Chitosan nanoparticles (100.0 ± 6.7 nm)	Cytoprotective autophagy	ROS generation	-	Hela cells and SMMC-7721 cells	[26]
Selenium nanoparticles (60 nm)	Activation of early autophagy but inhibition of late autophagy	Promoting apoptosis	Laminarin polysaccharides	HepG2 cells	[27]
Cuprous oxide NPs (200 nm)	ERK-dependent autophagy	ROS-mediated apoptosis	-	T24, J82, 5637, and UMUC3	[28]

**Table 3 pharmaceutics-14-02275-t003:** Some of the main proteins involved in mitophagy other than PINK1 and Parkin.

Protein	Role in Mitophagy	References
Beclin-1	Tumor suppressor protein actively involved in autophagy	[53]
X-Box Binding Protein 1 (XBP1)	Protein released upon oxidative stress that can induce autophagy in cancer cells via JNK activation and eIF2α phosphorylation	[101]
C/EBP homologous protein (CHOP)	Transcription factor required for the initiation of autophagy	[102]
Microtubule-associated protein 1A/1B-light chain 3 (LC3)	Conjugates with phosphatidylethanolamine to form LC3-phosphatidylethanolamine conjugate (LC3-II). It is recruited to the autophagosome membrane to assist degradation by lysosomes	[103]
Autophagy-related 3 (ATG3), Autophagy-related 12 (ATG12)	Proteins that are necessary to induce autophagy	[104][
Nitrogen permease regulator-like 2 (NPRL2)	Repress the mTOR signaling pathway to activate the process of autophagy and suppress apoptosis	[105]
BCL2 and adenovirus E1B19-kDa-interacting protein3 (BNIP3)	Mediate elimination of mitochondria by initiating LC3-dependent mitophagy, promote mitophagy by suppressing PINK1 cleavage	[106]
FUN14domain-containing protein1 (FUNDC1)	Essential role in mitochondrial quality control by mediating mitochondrial clearance by transducing hypoxia signals	[107]
Presenilin-associated rhomboid-like protein (PARL)	Regulator of PINK1 and Parkin	[108]

## Data Availability

Not applicable.

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
