# Peer review of "Mitophagy Induced by Metal Nanoparticles for Cancer Treatment"

_pharmaceutics, 2022, doi:10.3390/pharmaceutics14112275_

Round 1

Reviewer 1 Report

The present review article discusses the role of metal nanoparticles in the mitophagy process and how they can contribute to the restoration of this impaired process in cancer cells. The manuscript presented some critical and elaborated views of the topic. The topic is exciting and some suggestions are enclosed with comments/questions to revise this manuscript before publication.

-          -In general, in section 2 referring to the examples with some nanoparticles, the authors could explain a little about the mechanism, especially in gold nanoparticles, where the authors could complete this part by introducing the work of doi: 10.2147/IJN.S129274.

-          In figure 1 in some labels the letters seem out of focus

-          In the first paragraph of section 2 the line spacing or the letter are different

Author Response

Reviewer #1: The present review article discusses the role of metal nanoparticles in the mitophagy process and how they can contribute to the restoration of this impaired process in cancer cells. The manuscript presented some critical and elaborated views of the topic. The topic is exciting and some suggestions are enclosed with comments/questions to revise this manuscript before publication.

Response to Reviewer # 1:

We would like to thank the reviewer for the valuable and useful comments, we have revised the manuscript by addressing all the suggestions and comments to the best of our ability.

Comments to the author and response from the author:

Comment #1: In general, in section 2 referring to the examples with some nanoparticles, the authors could explain a little about the mechanism, especially in gold nanoparticles, where the authors could complete this part by introducing the work of doi: 10.2147/IJN.S129274.

Response #1: Thank you. The said reference was added to the manuscript by explaining about the role of gold nanoparticle in inducing mitochondrial dysfunction in NSCLC cell line involving the recruitment of dynamin-related protein 1 (Drp1), mitochondrial dysfunctions, and enhanced induction of autophagy. The changes are highlighted in the edited manuscript.  

Comment #2:  In figure 1 in some labels the letters seem out of focus

Response #2: The figure was re-drawn using BioRender app. The maximum clarity was achieved.

Comment #3: In the first paragraph of section 2 the line spacing or the letter are different

Response #3: Thank you for pointing out the mistake, we have corrected the editing mistake.

Reviewer 2 Report

In this review, Mundekkad and Cho review the use of nanoparticles inducing mitophagy in cancer. The review is clear but remains a bit superficial in terms of how to discern the main mechanism of action of these nanoparticles, how to achieve specificity and describe their composition. More information should be included to differentiate and discriminate the importance regarding other written reviews using nanoparticles to induce other types of autophagy processes and the importance of this type of targeting for cancer treatment vs. other strategies.

Major points:

1) Other reviews regarding the use of nanoparticles targeting autophagy have been written. The authors should compare and check the difference between targeting autophagy and mitophagy. Is it possible to differentiate between both strategies? Why one pathway vs the other one is essential or not? Describe strategies for generation nanoparticles targeting mitophagy and not other types of autophagy. Is it possible to select/Discriminate cells undergoing any of them or selectively induce any of them? What has been done in this sense, and do the authors want to review? Please, include this information in the text.

2) The authors should include a section regarding the type of nanoparticle composition (whole composition besides the difference in the loaded metal) and their size. that has been described to be used in cancer to induce mitophagy. Also, The authors assumed that all nanoparticles could cross the plasma membrane barrier, but this might not be something general and it is unclear how it accounts. The authors should summarize the type of used coating to target some cancer cells and not other cells specifically

3) The author should explain the difference between different metal-based nanoparticles and their mode of action. If there is any difference, they should explain the reason for that behavior.

4) The authors should summarize with a scheme the molecular pathways implicated in mitophagy, the effect of each type of nanoparticle effect on the specific targets described in section 4, and how this differential effect can be achieved if accounts.

5) Section 5 is too long in the review. Many things can be done in the future. This might be summarized in one paragraph at the end of the review, as future perspectives

Author Response

Response to Reviewer # 2:

We would like to thank the reviewer for the valuable and useful comments, we have revised the manuscript by addressing all the suggestions and comments to the best of our ability.

In this review, Mundekkad and Cho review the use of nanoparticles inducing mitophagy in cancer. The review is clear but remains a bit superficial in terms of how to discern the main mechanism of action of these nanoparticles, how to achieve specificity and describe their composition. More information should be included to differentiate and discriminate the importance regarding other written reviews using nanoparticles to induce other types of autophagy processes and the importance of this type of targeting for cancer treatment vs. other strategies.

Response – Thank you for your comments. We have taken your points, including to differentiate and discriminate the importance regarding other written reviews using nanoparticles to induce other types of autophagy processes, the main mechanism of action of these nanoparticles is explained in Table 2

Section 3 discusses how to achieve specificity

Section 2 and Table 2 discusses the probable mechanisms of action of nanoparticle. 

Major points:

  • Other reviews regarding the use of nanoparticles targeting autophagy have been written. The authors should compare and check the difference between targeting autophagy and mitophagy.

Response – Thank you for your comments, the differences and similarities between the two processes are included in Table 1

Is it possible to differentiate between both strategies?

Response - It is possible to differentiate between the two processes by understanding the target molecules and the proteins involved in the processes as explained in Table 1.

Why one pathway vs the other one is essential or not?

Response - One pathway is not more essential than the other pathway. Both are important for regular biogenesis and during disease conditions. Since mitophagy is the third type of autophagy (macroautophagy), we cannot say that one is important than the other per se. Both are operational in different situations and equally important for homeostasis.

Describe strategies for generation nanoparticles targeting mitophagy and not other types of autophagy.

Response - There is no specific strategy to generate nanoparticle to target mitophagy using nanoparticles. It is dependent on the physio-pathological condition of cellular microenvironment that the nanoparticle induce autophagy or mitophagy.  However, the nanoparticles can be utilized as nanocarriers to target the specific mitochondrial proteins which is explained in introduction as follows

‘Some of these studies used nanoparticles as effective drug carriers as they are very responsive to photosensitizers, radiosensitizers, and theranostic agents. They can target the energy machinery of tumor cells and effectively manipulate the underlying functional mechanisms in tumor cells’.

 Is it possible to select/Discriminate cells undergoing any of them or selectively induce any of them? What has been done in this sense, and do the authors want to review? Please, include this information in the text.

Response – Thank you for the thought-inducing question. According to the references, there is no way to select or discriminate the cells and direct them to induce mitophagy. However,  nanoparticles that have specificity towards certain receptors on the mitochondrial membrane can induce targeted mitophagy (explained in Section 2 - Selectivity of mitophagy by nanoparticles. The active targeting is in fact a way to direct the nanoparticles towards cells to induce mitophagy. This is further explained in the section mentioning active and passive targeting. And the specificity of nanoparticles to the receptors and proteins on the surface of mitochondria makes sure that that the nanoparticles are designed to induce mitophagy and not any other type of programmed cell death.

The authors should compare and check the difference between targeting autophagy and mitophagy. Is it possible to differentiate between both strategies?

Response -The differences are enlisted in Table 1. It is possible to differentiate between the two strategies based on the target molecules and proteins involved.

Table 1. Differences and similarities between autophagy and mitophagy

Autophagy

Mitophagy

Type

General form of degradation of cellular components including organelles

Specific degradation of mitochondria

Regulation

Dependent on the nutrient/energy/stress signals

Independent of the nutrient/energy/stress signals

Stimulus

Nutrient and energy stress, ER stress, pathogen-associated molecular patterns (PAMPs), danger-associated molecular patterns (DAMPs), hypoxia, redox stress, mitochondrial damage

Mitochondrial membrane depolarization, changes in cytosolic pH

Malleability

Malleable as it can degrade specific targets, entire organelles, and large portions of cytoplasm

Not malleable at all as it degrade only mitochondria

Substrate

SQSTM/p62

Mitophagy involves unique and additional substrate identification mechanisms, notably PTEN-induced kinase 1 (PINK1) and parkin RBR E3 ubiquitin protein ligase (PRKN)

Types

Three types –

1.       Chaperone-mediated autophagy (initiated by chaperone Hsc70 and recognizes one protein at a time),

2.       Microautophagy (initiated by invagination of lysosomal membranes. Lipid, protein or organelles can be degraded) and

3.       Macroautophagy (double membraned organelles are degraded)

Only one type

1.       where only mitochondria are degraded

Methods of detection

1.       Conventional electron microscopy to detect the autophagosome number and autophagic flux

2.       Fluorescence microscopy to count the average number of punctate structures per cell (puncta formation assay)

3.       Immunoblotting to detect the conversion of cytosolic form of LC3 (LC3-1) to its membrane-bound lipidated form (LC3-II)

4.       Flow cytometry to detect the degradation of autophagy-selective substrates

1.       Electron microscopy

2.       Fluorescence microscopy to detect co-localization of mitochondria with autophagosomes or lysosomes

3.       Western blotting to measure the degradation of mitochondrial proteins (like LC3)

4.       Fluorescent protein-tagged assays like MitoTimer, mt-Keima, and mito-QC

Similarities

Activation of similar molecular machinery (i.e., BECN1, ULK, LC3)

Generation of ROS molecules can act as a trigger for both processes

Both the processes results in cellular damage and apoptosis

Essential for quality control of the cells defense mechanism

Both the processes are forms of degradation mechanisms rather than cell death mechanism (unlike apoptosis). 

Included the explanation as below (lines 205 – 219)

Mitochondrial targeting can be active or passive. Active targeting is achieved by surface functionalization of the nanoparticles with mitochondria-specific ligands. These ligands could be special moieties that can either be loaded to the drug or can be standalone. Ligands like natural products (glycyrrhetinic acid), mitochondrial peptides, ?-tocopherol, etc. can be used for active targeting of nanoparticles to the mitochondrial membranes. However, the limitations like immunogenicity, high production costs, complexities during synthesis, off-target toxicity, prolonged blood circulation period, and delay in clearance from the biological system must be taken into account when designing the proper nanoparticle for active targeting. Passive targeting depends on physiological and chemical microenvironment of the mitochondria. pH, surface change, potential difference between both the membranes, surface functionalization, etc.  are factors affecting passive targeting of nanoparticles to mitochondria. This offers an advantage compared to active targeting as flexibility of nanoparticles allows the manipulation of their physical properties according to the requirements. But the disadvantage is the possibility of the formation of nanoparticle aggregates resulting in rapid clearance from the system.

The authors should include a section regarding the type of nanoparticle composition (whole composition besides the difference in the loaded metal) and their size. that has been described to be used in cancer to induce mitophagy.

Response – Thank you for the suggestion. The details are included in Table 2.

Table 2. Autophagy induced by different nanoparticles

Nanoparticle

(Size)

Mode of cellular

recycling

Mechanism/outcome

Conjugation

In vitro/in vivo model

Reference

Polydopamine nanoparticle

(101.96 ± 6.70 nm)

Autophagy

Photothermal cell killing

Beclin 1- derived peptide (Beclin 1), polyethylene glycol (PEG) and cyclic Arg-Gly-Asp (RGD) peptides (PPBR)

NIH3T3 cells, HeLa cells

[9]

AuNPs

(15 nm)

Autophagy

ROS generation by cellular uptake

Poly (acryloyl-L,D) and racemic valine (PAV)

MDA-MB-231 cells

[10]

Autophagy Cascade Amplification NPs (Self-assembled peptide-cholesterol monomers)

(150 nm)

Autophagy

Overactivated autophagy and enhanced tumor antigen processing

Oxaliplatin prodrug (HA-OXA)

CT26 tumor-bearing mice

[11]

ZnO NP

(300 nm)

Autophagy

ROS-mediated enhanced tumor chemotherapy by overstimulated autophagy

Bare

P-gp-mediated multidrug resistant human breast cancer cells (MCF-7/ADR cells),  BALB/c mice

[12]

Nickel oxide NPs

(24.05 ± 2.9 nm)

Autophagy

Oxidative stress, JNK activation

Bare

HeLa

[13]

Silver NPs

(78 nm)

Autophagy

Lysosome injury and cell hypoxia

Bare

Prostate cancer cell line (PC‐3)

[14]

Silver NPs

(11 – 23 nm)

Enhanced autophagy

Inhibition of NLRP3 inflammasome

Bare

THP-1 cells, AMJ-13 cells , HBL cells

[15]

Gold NPs

(60.00 ± 4.24 nm)

Autophagy

ROS-mediated cell death

Bare

Human ovarian adenocarcinoma cells (SKOV-3)

[16]

Hollow mesoporous titanium dioxide nanoparticles (HMTNPs)

(∼100 nm)

Escape from macrophage phagocytosis

Sonodynamic Therapy

Hydroxychloroquine sulphate (HCQ)

MCF-7, MDA-MB-231, HepG2, NIH3T3,

[17]

Copper oxide NPs

(5.4 ± 1.2 nm)

Destructive autophagy

Enhanced radio sensitizing effect

Bare

MCF-7

[18]

ZIF-82-PVP nanocrystals

(~240 nm)

Autophagy

Apoptosis promoted by X-ray induced nitrosative stress

Conjugation with PVP

MDA-MB-231, 4T1 and Panc-1 cells

[19]

Fe3O4 NP

(26.3 ± 4.42 nm)

Autophagy

ROS-mediated NF-κB and TGF-β signaling pathway activation

Polyethyleneimine

HeLa

[20]

Branched Au-Ag NPs

(~200 nm)

Autophagy

Photothermal toxicity

Polydopamine-coated

Human bladder cancer cells (T24 cells)

[21]

TiO2 NPs

(20 - 30 nm)

Autophagy blockage

Cytotoxicity and apoptosis induction with enhanced chemotherapeutic effect

5-fluorouracil

Human AGS gastric cancer cells

[22]

Au NPs

(42.6 ± 5.3 nm)

Autophagy

Immunogenic cancer cell death

Polydopamine

MCF-7 and MDA-MB231

[23]

Copper-palladium alloy tetrapod nanoparticle

(~50 nm)

Pro-survival autophagy

TNP-1-mediated photothermal therapy

-

Triple-negative (4T1), drug-resistant (MCF-7/MDR) and patient-derived breast cancer models

[24]

Spheroid fluorescent polystyrene nanoparticles (PS-NPs)

(30 nm)

Selective autophagy

Inhibition of autophagosome formation and rescued ATG4-mediated autophagy

Functionalized with Amino groups

OAW42 cells

[25]

Chitosan nanoparticles

(100.0 ± 6.7 nm)

Cytoprotective autophagy

ROS generation

-

Hela cells and SMMC-7721 cells

[26]

Selenium nanoparticles

(60 nm)

Activation of early autophagy  but inhibition of late autophagy

Promoting apoptosis

Laminarin polysaccharides

HepG2 cells

[27]

Cuprous oxide NPs

(200 nm)

ERK-dependent autophagy

ROS-mediated apoptosis

-

T24, J82, 5637, and UMUC3

[28]

The authors assumed that all nanoparticles could cross the plasma membrane barrier, but this might not be something general and it is unclear how it accounts.

Response – Thank you, the explanation for nanoparticle crossing barriers is included in Section 3.

It is essentially due to the physiochemical and biological advantages of nanoparticles that they are able to cross the biological barriers. Nanoparticles are known to have enhanced permeability and retention effect (EPR) that help them to easily accumulate in the permeable vasculature. The EPR effect along with the ability of nanoparticles to reach specific location to release drugs in a controlled mode can enhance their therapeutic index.      

3) The author should explain the difference between different metal-based nanoparticles and their mode of action. If there is any difference, they should explain the reason for that behavior.

Response –  Included the following section

Different metal-based nanoparticles and their mode of action

Metal-based nanoparticles are used extensively in medical field as drugs and imaging agents [37]. Understanding the basic mechanism of action of these nanoparticles will help in designing anticancer drugs for therapy and detection purposes. One of the ways by which the nanoparticles exert their therapeutic effect is the covalent bonding with biomolecules. The covalent binding of metal-based nanoparticles with other biomolecules is responsible for the extensive ligand exchange chemistry of the drug. The ligand exchange property of nanoparticles helps to understand and interpret the molecular events at atomic levels so as to study the variations in property of the drug. The formation of covalent amide bond with carboxylic acid on the surface of mesoporous silica nanoparticles, in synergy with zinc oxide quantum dots, was responsible for the inherent anticancer properties. Induction of mitophagy was found to be associated with the anticancer property triggered by excessive ROS formation [38]. The redox balance is important for cellular homeostasis that regulates a plethora of biological processes deciding the physiological well-being of the cell. Zinc oxide nanoparticles were found to induce ROS-mediated autophagy in CAL 27 oral cancer cell lines [39]. PINK1/Parkin-mediated mitophagy was reported as the basis for the anticancer activity of the metal oxide nanoparticle. The high expression of ROS in cancer cells results in the oxidation of cell components resulting in the loss of cell function and triggers autophagy. HC11 cells, treated with silver nanoparticles increased the expression of proteins associated with oxidative stress [40]. Hemeoxygnase-1 (HO-1), Kelch-like ECH-associated protein 1 (Kaep1), BTB and CNC homology proteins 1 (Bach1) and nuclear factor erythroid related factor 2 (Nrf2) were found to increase in the cells treated with the nanoparticles.

Another mode of action of nanoparticles it the activation of immune system by phototherapy. Metal-based nanoparticles are excellent photosensitizers; they can be easily sensitized even by low intensity light sources as in photodynamic therapy (PDT) and the irradiation creates ROS in cancer cells that can affect the mitochondrial membrane potential to induce mitophagy. In photothermal therapy (PTT), the nanoparticles sensitized with intense light and electromagnetic waves can instigate the expression of heat shock proteins (HSP) on the cell surface that simultaneously leads to the release of cytokines and other inflammatory regulators [2]. The immune system of the cells’ defense mechanism is activated upon the release of antigens in photo immune therapy (PIT). The interventional PDT and PTT can trigger signaling molecules that act on the mitochondrial proteins directing the mitochondria for destruction by mitophagy. Iron oxide nanoparticles were directed to induce mitophagy in MCF-7 cell lines through photothermal effect. When the photosensitized nanoparticles enter the cancer cells, they form aggregates in lysosomes and exocytosis is inhibited [41].

4) The authors should summarize with a scheme the molecular pathways implicated in mitophagy, the effect of each type of nanoparticle effect on the specific targets described in section 4, and how this differential effect can be achieved if accounts.

Response - Additional figure have been included in the manuscript describing the molecular pathways implicated in mitophagy induced by nanoparticles (Fig. 3). Most of the nanoparticles follow the events that are dependent on Pink1/parkin pathway with regular cross-talks between the various molecular mechanisms. Other pathways that are involved are also interconnected by common players like LC3-II and formation of autophagosomes. Further, Fig. 4 also describes the crosstalk between different signaling pathways in nanoparticle-induced mitophagy 

  • Section 5 is too long in the review. Many things can be done in the future. This might be summarized in one paragraph at the end of the review, as future perspectives

Response -  Thank you for the suggestion. The given suggestions are followed, and the section is changed as suggested by the reviewer and summarized in one paragraph.  

Round 2

Reviewer 2 Report

All my questions were answered in detail by the authors. I do not have more questions.